 eLIFE

# Diagnostically relevant facial gestalt information from ordinary photos

Quentin Ferry[1,2], Julia Steinberg[2,3], Caleb Webber[2], David R FitzPatrick[4], Chris P Ponting[2], Andrew Zisserman[1]*, Christoffer Nellåker[2]*

[1]Department of Engineering Science, University of Oxford, Oxford, United Kingdom; [2]Medical Research Council Functional Genomics Unit, Department of Physiology, Anatomy and Genetics, University of Oxford, Oxford, United Kingdom; [3]The Wellcome Trust Centre for Human Genetics, University of Oxford, Oxford, United Kingdom; [4]Medical Research Council Human Genetics Unit, Institute of Genetics and Molecular Medicine, Edinburgh, United Kingdom

**Abstract** Craniofacial characteristics are highly informative for clinical geneticists when diagnosing genetic diseases. As a first step towards the high-throughput diagnosis of ultra-rare developmental diseases we introduce an automatic approach that implements recent developments in computer vision. This algorithm extracts phenotypic information from ordinary non-clinical photographs and, using machine learning, models human facial dysmorphisms in a multidimensional 'Clinical Face Phenotype Space'. The space locates patients in the context of known syndromes and thereby facilitates the generation of diagnostic hypotheses. Consequently, the approach will aid clinicians by greatly narrowing (by 27.6-fold) the search space of potential diagnoses for patients with suspected developmental disorders. Furthermore, this Clinical Face Phenotype Space allows the clustering of patients by phenotype even when no known syndrome diagnosis exists, thereby aiding disease identification. We demonstrate that this approach provides a novel method for inferring causative genetic variants from clinical sequencing data through functional genetic pathway comparisons.

*For correspondence: az@robots.ox.ac.uk (AZ); christoffer.nellaker@dpag.ox.ac.uk (CN)

## Introduction

Genetic disorders affect almost 8% of people (*Baird et al., 1988*), about a third of whom will have symptoms that greatly reduce their quality of life. While there are over 7000 known inherited disorders, only a minority of patients with a suspected developmental disorder receive a clinical, let alone a genetic, diagnosis (*Hart and Hart, 2009*). A genetic diagnosis allows more specific therapeutic interventions to be investigated and can aid the identification of primary vs secondary symptoms.

The introduction of whole genome and exome sequencing into modern clinical medicine will be instrumental in raising the current low rate of genetic diagnoses for ultra-rare diseases. Nevertheless, tools to accurately assign functional and disease relevance to sequence variants are substantially lacking. Projects that apply next generation sequencing to patients in clinical settings fail to report genetic diagnoses for approximately 80% of cases (*de Ligt et al., 2012*). The difficulty lies in identifying the causal variant in an individual patient: even when ignoring experimental error, each individual carries approximately 4 million differences, in the case of whole genome sequencing, relative to the reference genome. Computational analyses currently are able only to interpret the ~2500 variants that alter protein sequence at evolutionarily conserved positions and ~400 very rare variants that are likely to be causal for pathogenic processes (*Abecasis et al., 2012*). Notably, of the ~10% of the genome that is functional all except the 1.2% that is protein-coding is often disregarded (*Weischenfeldt et al., 2013*).

**eLife digest** Rare genetic disorders affect around 8% of people, many of whom live with symptoms that greatly reduce their quality of life. Genetic diagnoses can provide doctors with information that cannot be obtained by assessing clinical symptoms, and this allows them to select more suitable treatments for patients. However, only a minority of patients currently receive a genetic diagnosis.

Alterations in the face and skull are present in 30–40% of genetic disorders, and these alterations can help doctors to identify certain disorders, such as Down's syndrome or Fragile X. Extending this approach, Ferry et al. trained a computer-based model to identify the patterns of facial abnormalities associated with different genetic disorders. The model compares data extracted from a photograph of the patient's face with data on the facial characteristics of 91 disorders, and then provides a list of the most likely diagnoses for that individual. The model used 36 points to describe the space, including 7 for the jaw, 6 for the mouth, 7 for the nose, 8 for the eyes and 8 for the brow.

This approach of Ferry et al. has three advantages. First, it provides clinicians with information that can aid their diagnosis of a rare genetic disorder. Second, it can narrow down the range of possible disorders for patients who have the same ultra-rare disorder, even if that disorder is currently unknown. Third, it can identify groups of patients who can have their genomes sequenced in order to identify the genetic variants that are associated with specific disorders.

The work by Ferry et al. lays out the basic principles for automated approaches to analyze the shape of the face and skull. The next challenge is to integrate photos with genetic data for use in clinical settings.

Therefore, the prediction of causal inherited variants in an individual can result in high false positive and high false negative rates.

The most powerful approach to associate a particular gene with an ultra-rare disease is to identify multiple unrelated individuals with the disorder whose genomes harbor deleterious alleles in a shared gene, regulatory element or pathway (*Schuurs-Hoeijmakers et al., 2012*). However, this approach relies on at least two individuals with the same disorder being available for comparison, an unlikely event given that these two individuals are selected for comparison from the roughly 100 million people affected by rare developmental disorders (prevalence of less than 2 per 100,000 around the world) (*Orphanet, 2013*). For the past 65 years, clinical geneticists have studied, diagnosed, and characterized developmental disorders on the basis of common characteristics among patients (*Rimoin and Hirschhorn, 2004*). When a given causal variant is ultra-rare, however, this presents substantial difficulties. Consequently, to realize the full potential of next generation sequencing in clinical diagnostics, phenotypic characterization must also become correspondingly high throughput and sensitive (*Hennekam and Biesecker, 2012*).

The facial gestalt provides valuable information to identify similarities between patients because 30–40% of genetic disorders manifest craniofacial abnormalities (*Hart and Hart, 2009*). The utility of computer vision for diagnosis and phenotyping of dysmorphic disorders has been explored previously by several groups and with varying approaches (*Loos et al., 2003*; *Hammond et al., 2005*; *Hammond, 2007*; *Boehringer et al., 2006*; *Dalal and Phadke, 2007*; *Vollmar et al., 2008*; *Boehringer et al., 2011*, reviewed in *Hammond and Suttie, 2012*; *Baynam et al., 2013*). The computational analysis of facial morphology using 3D imaging has been applied to conditions such as fetal alcohol syndrome (*Suttie et al., 2013*), schizophrenia (*Buckley et al., 2005*; *Hennessy et al., 2006*, *2007*) and autism (*Aldridge et al., 2011*). While 3D imaging studies have shown high discriminatory power in terms of classification they have relied on specialized imaging equipment and patient cooperation. Previous work with 2D images has relied on manual annotation of images, controlling lighting, pose and expression to allow consistent analyses. These factors greatly limit the availability, and ultimately the potential widespread clinical utility of such approaches.

We have adopted a complementary approach that takes advantage of the wealth of data available for human faces, an indirect result of the ubiquitous availability of cameras. To do so we provide a new representation ('Clinical Face Phenotype Space'), which is an application of computer vision and machine learning algorithms for analyzing craniofacial dysmorphisms from ordinary photographs.

We have ensured that Clinical Face Phenotype Space is robust to spurious variations such as lighting, pose, and image quality which would otherwise bias analyses. The approach is fully automated and provides objective and consistent computational descriptions of facial gestalt. Our method both greatly narrows the search space for investigating known disorders and will increase the power of inferring causative variants in previously unidentified genetic disease.

## Results

We sought to construct a database of patient photos within which faces would be automatically identified and their key features annotated. Our intent was to build a model of dysmorphic variation from a set of syndromes that, additionally, would be able to cluster syndromes not used in model training. Our schema by which a patient photo is automatically analyzed within the context of Clinical Face Phenotype Space is provided in *Figure 1A*.

### Image database composition

We first collected a database of 2878 images, including 1515 healthy controls and 1363 pictures for eight known developmental disorders from publically available sources across the internet (*Table 1*, references for image sources are available from *Supplementary file 1*). Manual checks were performed to exclude images where the face or an eye was not clearly visible, or where an expert clinician (DRF) could not verify the diagnosis. Manual annotation of facial features points was performed on all images to allow training and testing of an automated annotation algorithm. These initial requirements for manual intervention are dispensed with in the final automatic algorithm (see below).

### Computer vision algorithms

We proceeded to train a computer vision algorithm for automatic annotation of 36 feature points of interest across the face (*Figure 1A*). Our approach takes advantage of a variety of facial detection algorithms (OpenCV [*Bradski, 2000*], Viola Jones [*Viola and Jones, 2001*] and Everingham [*Everingham et al., 2009*]) and custom learning (consensus of exemplars [*Belhumeur et al., 2011*]) to accurately place feature points on a given face ('Materials and methods'). Across all images in our database, manual checking found that our algorithm detected and annotated 99.5% of tested faces correctly with accuracies in the range 6–60% of the width of an eye (individual feature point accuracies are provided in *Figure 1—figure supplement 1*).

We used an Active Appearance Model ('Materials and methods') to calculate an average face within any set of images, representing consistent shape and appearance features within the group (*Figure 1B* and animated morphs in *Figure 2*). The average faces for each syndrome show that the algorithm effectively captures characteristic features of dysmorphic syndromes (*Figure 2—figure supplement 1*). For each feature point, the algorithm extracts a feature vector describing appearance of the surrounding patch. The algorithm then constructs a feature vector describing shape based on the relative pairwise distances between all feature points ('Materials and methods'). We next sought to compare the syndrome relevant information content of the feature descriptors to previous studies (*Hammond et al., 2005*; *Boehringer et al., 2006*; *Hammond, 2007*; *Vollmar et al., 2008*). We found that classification analysis based on support vector machines provided similar accuracies to previous work, despite disparities in image variability (average classification accuracy 94.4%, see Figure 4—figure supplement 1, Figure 4—figure supplement 2 and 'Materials and methods').

It is important to emphasize that the analyzed images vary greatly, as there were minimal restrictions imposed on image selection placed by the two exclusion criteria (both eyes visible and diagnosis verified by DRF). Photos were analyzed irrespective of the subject's age, gender, facial expression or ethnicity or the background scenery. Principal component analysis (PCA) of facial descriptor vectors illustrates that the main sources of variation among images are indeed lighting, pose, and facial expression, rather than phenotypic features (*Figure 1—figure supplement 2*).

### Constructing a Clinical Face Phenotype Space with metric learning

We next performed Metric Learning using a Large Margin Nearest Neighbor (*Weinberger and Saul, 2009*) approach for the eight syndromes in the database. This approach linearly transformed the multidimensional space of PCA feature vectors to optimize the separation of syndromes: dimensions informative for dysmorphism phenotypes are expanded while uninformative dimensions are compressed (thus changing the relative importance for clustering). We denote the

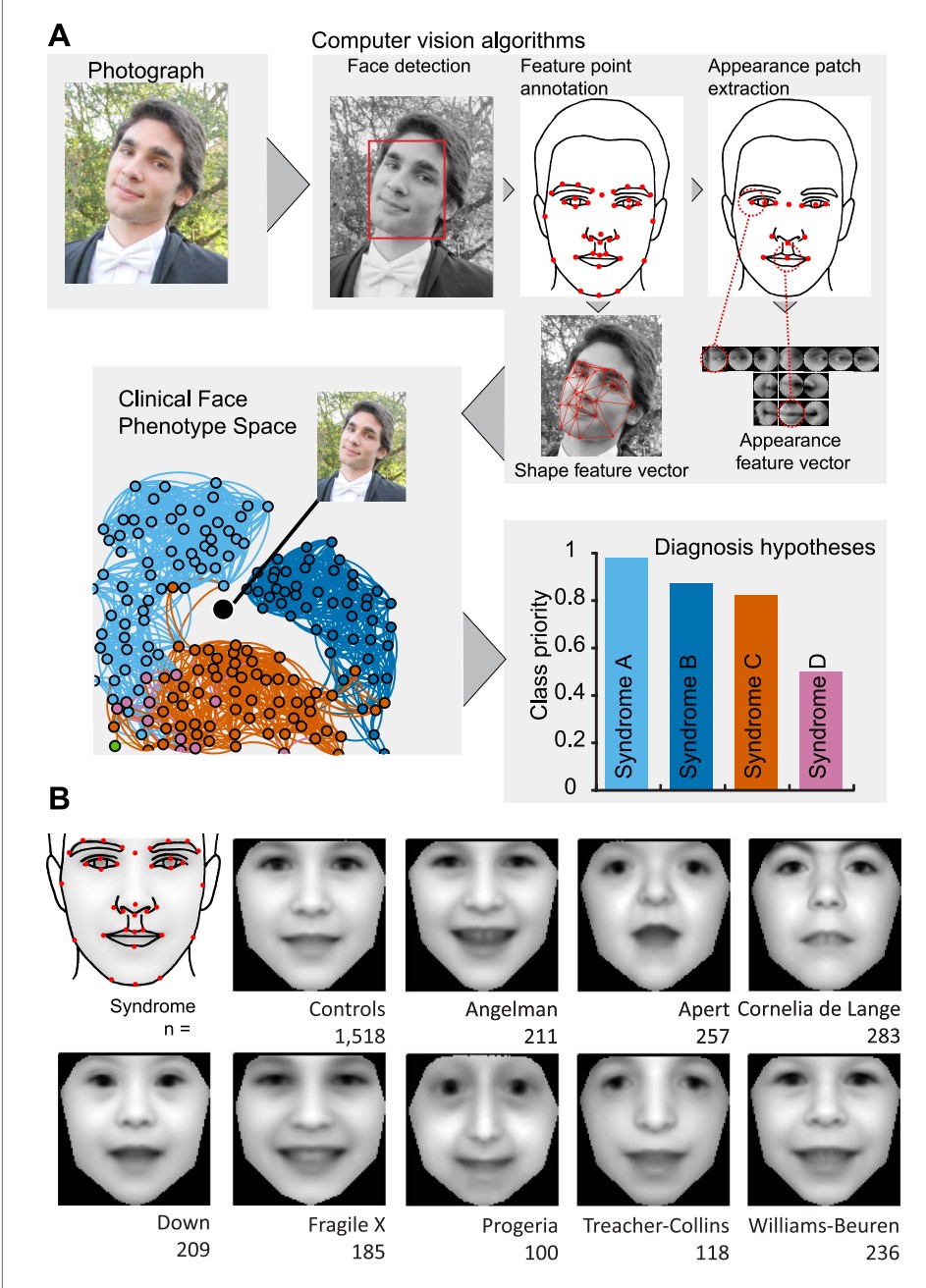

**Figure 1**. Overview of the computational approach and average faces of syndromes. (**A**) A photo is automatically analyzed to detect faces and feature points are placed using computer vision algorithms. Facial feature annotation points delineate the supra-orbital ridge (8 points), the eyes (mid points of the eyelids and eye canthi, 8 points), nose (nasion, tip, ala, subnasale and outer nares, 7 points), mouth (vermilion border lateral and vertical midpoints, 6 points) and the jaw (zygoma mandibular border, gonion, mental protrubance and chin midpoint, 7 points). Shape and Appearance feature vectors are then extracted based on feature points and these determine the photo's location in Clinical Face Phenotype Space (further details on feature points in *Figure 1—figure supplement 1*). This location is then analyzed in the context of existing points in Clinical Face Phenotype Space to extract phenotype similarities and diagnosis hypotheses (further details on Clinical Face Phenotype Space with simulation examples in *Figure 1—figure supplement 2*). (**B**) Average faces of syndromes in the database constructed using AAM models ('Materials and methods') and number of individuals which each average face represents. See online version of this manuscript for animated morphing images that show facial features differing between controls and syndromes (*Figure 2*).

*Figure 1. Continued on next page*

*Figure 1. Continued*
The following figure supplements are available for figure 1:
**Figure supplement 1**.
**Figure supplement 2**. Phenotypic vs spurious feature variation in Clinical Face Phenotype Space using simulated faces.

resulting transformed 270 dimensional space as 'Clinical Face Phenotype Space' (see 'Materials and methods').

Due to its design, Clinical Face Phenotype Space clusters patient faces based on diagnostically relevant phenotypic features, while tolerating spurious variation. Relative importance of spurious and phenotypic variation for clustering in Clinical Face Phenotype Space was tested using simulated faces ('Materials and methods'). For these faces feature dimensions that reflected known spurious variation such as lighting and head orientation were compressed and hence were of less relevance for clustering (*Figure 1—figure supplement 2*).

For the eight syndromes with which Clinical Face Phenotype Space was created, we performed tests with supervised learning and clustering. A kNN-classifier applied within Clinical Face Phenotype Space was able to correctly classify images with an accuracy of 99.5% using the leave-one-out method. However, to avoid biases introduced by training data size, we also assessed the improvements in clustering by measuring the search space reduction (hereafter referred to as the Clustering Improvement Factor or CIF, 'Materials and methods'). This estimates the factor by which the Clinical Face Phenotype Space improves the clustering of syndromes when compared with random chance (to 95% confidence). On average, the clustering of the eight syndromes within the database was improved by 11.0-fold (geometric mean of improved clustering, CIF range 9.1–23.5, maximum possible mean 12.5; *Figure 3*).

Next, we tested and confirmed our hypothesis that Clinical Face Phenotype Space could be generalized to dysmorphic syndromes that were not used in the training. We had access to 75 syndromes from the Gorlin collection (a kind gift curated and annotated by Professor Raoul Hennekam, Academic Medical Center, University of Amsterdam), which we supplemented with additional images of 22q11, Marfan and Sotos syndromes. Furthermore, we collected images of patients with verified genetic mutations in *PACS1* or in specific genes from the RAS/MEK pathway (*Supplementary file 1* references for image sources in 'Materials and methods'). The number of individuals within each syndrome varied between 2 and 223. The search space reduction was on average 27.6-fold better than random chance (CIF range 1.0–700.0, maximum possible average CIF was 150.0; *Figure 4A*). That is to say, that among 2754 patients' faces associated with any of 90 syndromes Clinical Face Phenotype Space makes it 27.6-fold easier to make the correct diagnosis. This demonstrates that Clinical Face Phenotype Space is an effective approach to the identification of multiple individuals sharing ultra-rare, previously undocumented, genetic disorders.

We proceeded to test if Clinical Face Phenotype Space recapitulates the modularity of genetic diseases, where clusters of phenotypically similar disorders reflect functional relationships among the genes involved (see *Oti and Brunner, 2007* for a review). We have shown that individuals with the same underlying genetic disease automatically cluster in Clinical Face Phenotype Space. We next tested whether disorders caused by mutations in different genes result in meaningful clusters in Clinical Face Phenotype Space. We selected disorders with a known genetic origin, using either gene associations from OMIM or publications describing the identification of causative genes (see 'Materials and methods'). For each pair of genes, the shortest path in a protein–protein interaction network was obtained from Dapple (*Rossin et al., 2011*), giving a protein interaction distance relevant to that gene pair. We compared genes underlying monogenic syndromes linked by 1, 2, or 3 path distances, with those with a path distance of 4 or that was unknown; unknown distances are those where no genes are associated with a syndrome, the syndrome is multigenic, or when DAPPLE has no known interaction documented, see 'Materials and methods'. For each pair of syndromes, an average Euclidean distance in Clinical Face Phenotype Space was calculated. The distance in Clinical Face Phenotype Space is significantly shorter between syndromes associated to genes with protein interaction distances of 1, 2, or 3 compared with syndromes with 4 or no known interactions ($p < 0.01$, $p < 0.05$ and $p < 0.001$ respectively, *Figure 5*). This demonstrates that the distance in Clinical Face Phenotype Space partly

**Table 1.** Composition of the database

| Syndrome | Nr images | Syndrome | Nr images |
|---|---|---|---|
| Public images online | | Published images | |
| Angelman | 205 | PACS1 | 2 |
| Apert | 203 | BRAF | 35 |
| Cornelia de Lange | 179 | CFC | 1 |
| Down | 199 | Costello | 10 |
| Fragile X | 164 | ERF | 5 |
| Progeria | 78 | HRAS | 5 |
| Treacher Collins | 103 | KRAS | 12 |
| Williams-Beuren | 232 | MAP2K1 | 5 |
| | | MAP2K2 | 4 |
| Controls | 1515 | MEK1 | 5 |
| | | NRAS | 2 |
| 22q11 | 8 | PTPN11 | 19 |
| Marfan | 18 | RAF1 | 9 |
| Sotos | 36 | SHOC2 | 8 |
| Turner | 12 | SOS1 | 30 |
| The Gorlin Collection | | | |
| Aarskog | 19 | Klippel-Trenaunay | 10 |
| Achondroplasia | 12 | Langer-Giedion | 14 |
| Alagille | 8 | Larsen | 11 |
| Albright | 7 | Lenz_Majewski | 17 |
| Angelman | 13 | Lymphedema-Lymphangiectasia-MR | 8 |
| Apert | 49 | Melnick_Needles | 17 |
| Beckwith-Wiedemann | 11 | Moebius | 9 |
| Bloom | 9 | Muenke | 15 |
| BOF | 15 | Myotonicdystrophy | 9 |
| Cartilagehair | 13 | Neurofibromatosis | 7 |
| CHARGE | 12 | Noonan | 29 |
| Cherubism | 20 | OAVdysplasia | 18 |
| CleidoCranialdysostosis | 13 | ODD | 21 |
| Coffin-Lowry | 20 | OFCD | 10 |
| Costello | 9 | OFD | 18 |
| CriduChat | 17 | OPD | 31 |
| Crouzon | 16 | Osteopetrosis | 2 |
| Crouzonodermoskeletal | 5 | Osteosclerosis | 5 |
| Cutislaxa | 11 | Otodental | 2 |
| DeLange | 17 | Poland | 4 |
| Diastrophicdysplasia | 5 | Prader–Willi | 16 |
| Down | 8 | Progeria | 14 |
| Dubowitz | 12 | Proteus | 6 |
| Dyggve-Melchior-Clausen | 8 | Rieger | 4 |
| EEC | 6 | Rothmund-Thomson | 13 |
| Ehlers-Danlos | 17 | Rubinstein-Taybi | 8 |
| Ellis-vanCreveld | 3 | Saethre-Chotzen | 25 |

*Table 1. Continued on next page*

Table 1. Continued

| Syndrome | Nr images | Syndrome | Nr images |
|---|---|---|---|
| FG | 11 | Sclerosteosis | 4 |
| FragileX | 27 | SeckelMOD | 7 |
| Frontometaphysealdysplasia | 12 | SEDcongenita | 6 |
| Gorlin | 91 | Sotos | 16 |
| Gorlin_Chaudry_Moss | 13 | Stickler | 42 |
| Greig | 7 | TRP | 24 |
| Hallermann-Streiff | 9 | Waardenburg | 39 |
| Incontinentiapigmenti | 4 | Weaver | 13 |
| Kabuki | 25 | Williams-Beuren | 19 |
| Klippel-Feil | 3 | | |

recapitulates the functional relatedness of underlying developmental processes known to be disrupted in genetic diseases.

## Querying Clinical Face Phenotype Space

Clinical Face Phenotype Space can provide clinical phenotyping and clustering to known genetic disorders that is objective and high-throughput. The method is, however, neither sufficiently accurate nor intended to determine diagnosis, yet it can help to narrow the diagnostic search space in an unprejudiced manner. A clinician could easily photograph a patient and immediately obtain clinically useful diagnostic hypotheses and matching cases. To this end, we implemented two primary methods to automatically and objectively query Clinical Face Phenotype Space.

For any given image located in Clinical Face Phenotype Space, we obtain confidence ranked classifications to known disorders (see 'Materials and methods' and *Figure 4—figure supplement 4*). In addition, we objectively compare the image to others within the space. For any given query image, a probabilistic ranking of similar syndromes is obtained through nearest neighbor representation compared to random expectation of clustering among the 90 syndromes and 2754 faces. The classification confidence for a particular disorder depends on its location within the space, but also on the local densities of similar faces. We find that for the eight initial syndromes used to construct Clinical Face Phenotype Space, 93.1% (range 81.0–99.2%) are correctly classified as the top rank, cumulatively converging on 99.1% (95.8–100%) by the 20th rank (*Figure 4B*). Of syndromes not part of the Clinical Face Phenotype Space training, the classification accuracies positively correlated strongly with the number of instances in the database (*Figure 4B*). For the 20 syndromes where the database held 5 or fewer examples (*Table 1*), we classify on average 20.3% correctly by the 6th rank (exceeding 16.3-fold better than by chance alone).

For individuals with a suspected ultra-rare or an undocumented novel disorder, we developed a metric, p0p1, which assesses their similarity to others within Clinical Face Phenotype Space. The metric estimates the relative closeness of two faces given an average local density with the space: a p0p1 value exceeding 1 indicates a potentially new cluster, see 'Materials and methods'. The 2 PACS1 cases reported by *Schuurs-Hoeijmakers et al. (2012)* placed within Clinical Face Phenotype Space have a p0p1 value of 1.05 meaning that they are 5% closer to one another than the geometric mean of the distances to their 20 nearest neighbors. Taking into account that this is a local density estimate among 2754 faces in Clinical Face Phenotype Space, the search space to find them has been reduced ~690.4-fold (CIF, see 'Materials and methods').

The combination of syndrome clustering and de novo similarity metrics should aid the diagnosis of known syndromes and provides a means of clustering patients where no documented diagnosis exists.

## Discussion

We have developed our algorithm on normal-everyday 2D photographs and have focused on 36 facial feature points. Given the orders of magnitude lower dimensionality of our data as compared to a 3D

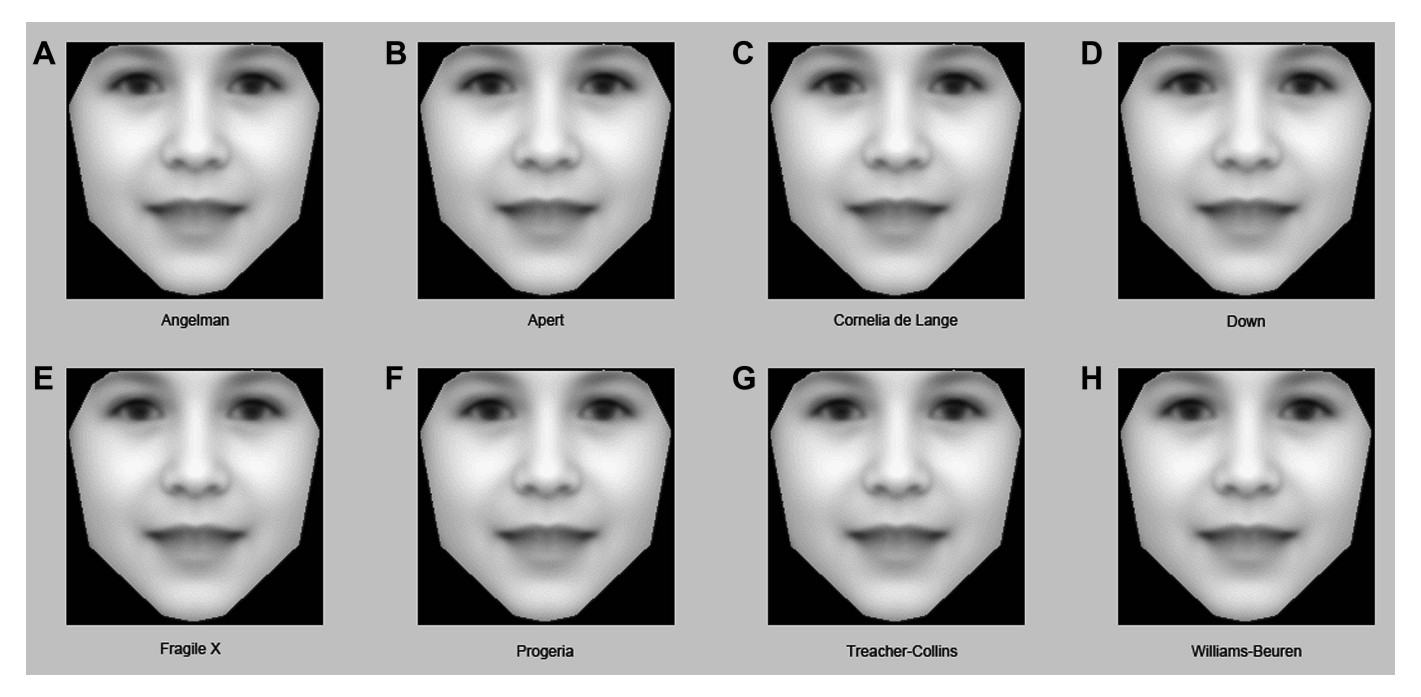

**Figure 2**. Animated morphs of average faces from controls to syndromes. (**A**) Angelman, (**B**) Apert, (**C**) Cornelia de Lange, (**D**) Down, (**E**) Fragile X, (**F**) Progeria, (**G**) Treacher-Collins, (**H**) Williams-Beuren. Delineation of syndrome gestalt relative to controls with distortion graphs in *Figure 2—figure supplement 1*.

The following figure supplements are available for figure 2:

**Figure supplement 1**. Distortion graphs representing the characteristic deformation of syndrome faces relative to the average control face.

imaging capture (*Hammond et al., 2005*), we were initially concerned that this would be insufficient to capture facial phenotypes. However, we then demonstrated that the approach is able to describe and discriminate between syndromes with a comparable accuracy to previous studies (*Loos et al., 2003*; *Hammond et al., 2005*; *Hammond, 2007*; *Boehringer et al., 2006*; *Dalal and Phadke, 2007*; *Vollmar et al., 2008*; *Boehringer et al., 2011*). The accessibility of normal 2D photographs (as opposed to 3D imaging) should outweigh any lower data resolution obtained from any one image and in future developments using multiple profile perspectives will allow 3D structure to be inferred. With accurate registration of a person's face from multiple images across time, from a family photo album for instance, it would capture not only the 3D structure but also the progression and development of dysmorphic gestalt. The automatic image analysis algorithm enables phenotypic metrics to be obtained with objective consistency from each image (*Figure 1*).

Clinical Face Phenotype Space was instantiated using eight syndromes that were well populated in our database so as to be robust against spurious variation. In doing so, it has become a generalizable model for craniofacial dysmorphic variation (*Figure 5*). The high fidelity of the current Clinical Face Phenotype Space (*Figure 3*) shows promise given that known deficiencies have yet to be addressed: (1) We used only single image examples of individuals. (2) The spectrum of phenotypes represented was limited. (3) The average image quality in the database was low. (4) The current 36 facial feature points only capture full frontal facial phenotypes, and thus miss valuable information from the full cranium and profile perspectives. Among the approaches that will be tested in future work are: increasing the number of feature points across the cranium, using profile images and taking advantage of multiple images of the same individual. Furthermore, we will be exploring performing explicit modelingmodeling of the 3D variation for 2D images (*Ramnath et al., 2008*), other types of feature descriptors, alternative metric learning and dimensionality reduction approaches (*Simonyan et al., 2013*). As Clinical Face Phenotype Space is developed and populated with more individuals, the predictive

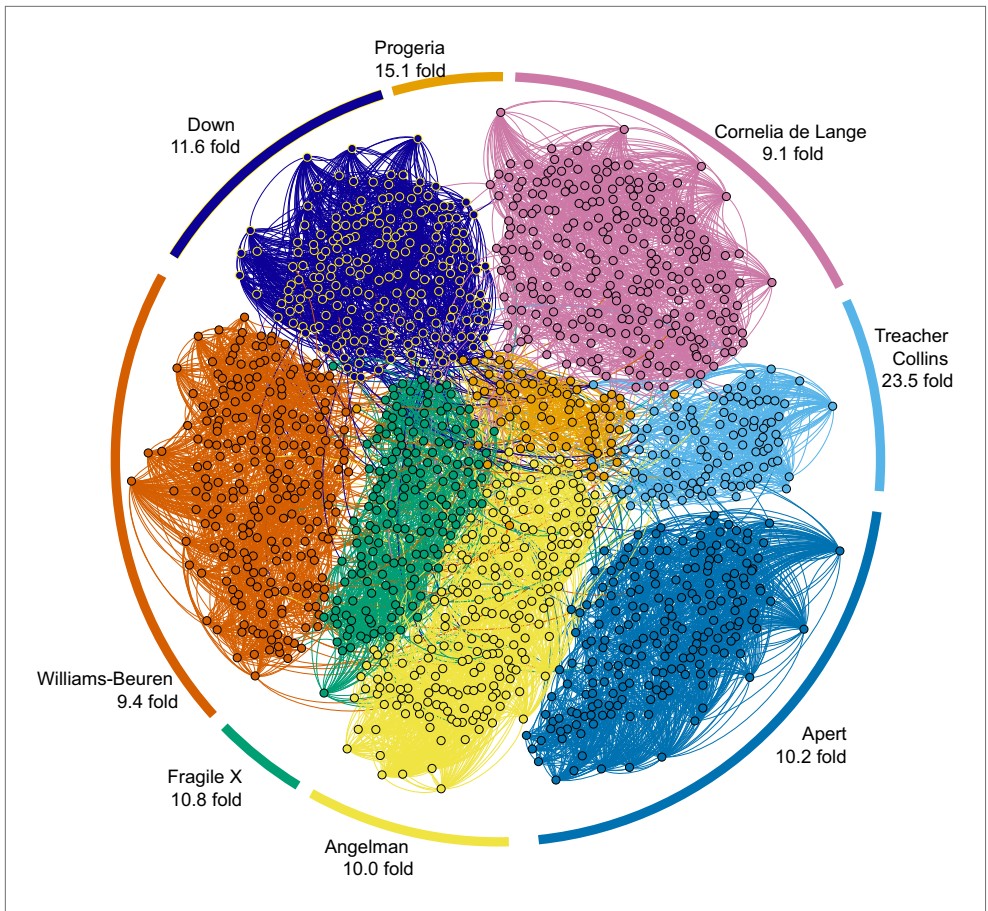

**Figure 3**. Clinical Face Phenotype Space enhances the separation of different dysmorphic syndromes. The graph shows a two dimensional representation of the full Clinical Face Phenotype Space, with links to the 10 nearest neighbors of each photo (circle) and photos placed with force-directed graphing. The Clustering Improvement Factor (CIF, fold better clustering than random expectation) estimate for each of the syndromes is shown along the periphery.

power to infer novel causative genetics would be expected to increase linearly until it asymptotically approaches a theoretical maximum.

There are three anticipated primary applications for Clinical Face Phenotype Space in a clinical setting: narrowing the search space for documented developmental disorders, identifying multiple people that share an ultra-rare genetic disorders and aiding the inference of causative variants in clinical genetic sequencing (*Figure 4*).

We envisage Clinical Face Phenotype Space becoming a standard tool to support clinical genetic counseling. Since any normal 2D image can be analyzed, this approach is available to any clinician worldwide with access to a camera and a computer. This can also reduce the need for patient inconvenience in a clinical setting because a family photo album could provide the required image(s). A photograph will enable automatic digital phenotyping, and its placement in Clinical Face Phenotype Space will provide an unbiased list of candidate clinical hypotheses (exemplified in *Figure 6*). We anticipate that future developments of Clinical Face Phenotype Space will also identify sub-phenotypes or comorbidities. Where no known genetic disease or variant can be assigned, Clinical Face Phenotype Space can identify other patients with phenotypic similarities empowering the identification of ultra-rare genetic disorders.

In summary, we have presented an algorithmic approach that provides a critical advance in applying computer vision and machine learning techniques as a tool for clinical geneticists. The conjunction of a computer vision and machine learning algorithm with Clinical Face Phenotype Space makes this

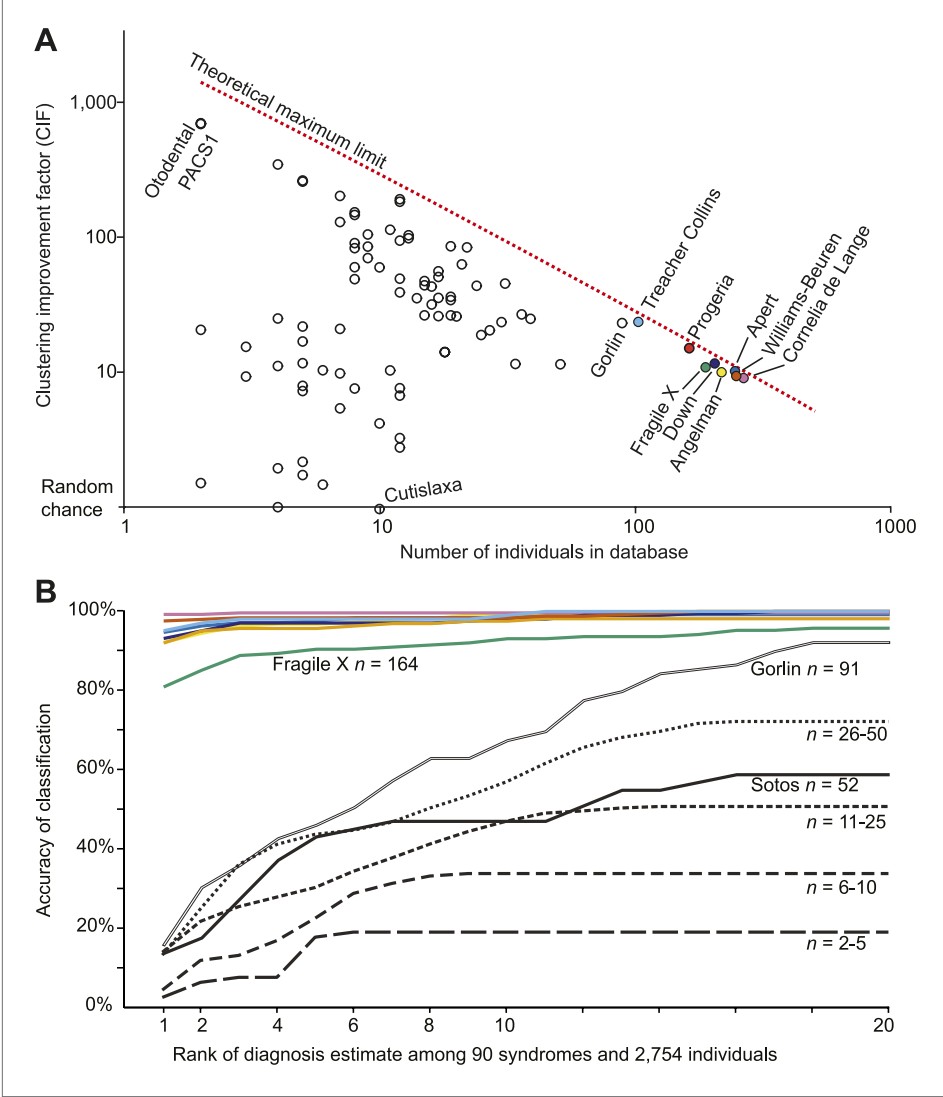

**Figure 4**. Clinical Face Phenotype Space is generalizable to dysmorphic syndromes that are absent from a training set. (**A**) Clustering Improvement Factor (CIF) estimates are plotted vs the number of individuals per syndrome grouping in the Gorlin collection or patients with similar genetic variant diagnoses. As expected, the stochastic variance in CIF is inversely proportional to the number of individuals available for sampling. The median CIF across all groups is 27.6-fold over what is expected by clustering syndromes randomly. That is to say, the CIF of a randomly placed set is 1. The maximum CIF is fixed by the total number of images in the database and by the cardinality of a syndrome set: the theoretical maximal CIF upper bound is plotted as a red dotted line. The CIF for the minimum and maximum, Cutislaxa syndrome and Otodental syndrome, were 1.0 and 700.0 respectively. (**B**) Average probabilistic classification accuracies of each individual face placed in Clinical Face Phenotype Space (class prioritization by 20 nearest neighbors weighted by prevalence in the database). The 8 initial syndromes used to train Clinical Face Phenotype Space are shown in color. For syndromes with fewer than 50 examples, accuracies were averaged across all syndromes binned by data set size (i.e., the average accuracy is shown for syndromes with 2–5, 6–10, 11–25, and 26–50 images in the database, *Supplementary file 1*). Classification accuracies increase proportional to the number of individuals with the syndrome present in the database. Accuracies using support vector machines with binary and forced choice classifications are shown in *Figure 4—figure supplement 1* and *Figure 4—figure supplement 2*. A simulation example of probabilistic querying of Clinical Face Phenotype Space is shown in *Figure 4—figure supplement 3*.

The following figure supplements are available for figure 4:

**Figure supplement 1**. SVM binary classification accuracies among the 8 syndromes in *Table 1*.

*Figure 4. Continued on next page*

*Figure 4. Continued*

**Figure supplement 2**. SVM forced choice classification accuracies among the 8 syndromes in *Table 1*.

**Figure supplement 3**. Simulated example illustrating the Clustering Improvement Factor.

**Figure supplement 4**. Simulated example of probabilistic querying of Clinical Face Phenotype Space.

approach high-throughput, automatic, objective, and broadly accessible with existing digital photography and computers. Our ongoing research has begun to apply the Clinical Face Phenotype Space approach within large clinical sequencing collaborations. Computer vision for aiding diagnosis of developmental disorders in clinical genetics will be tenable and broadly applicable in the near future.

## Materials and methods

### Database collection

We built a database of publically available or scientifically published pictures of patients collected across the internet. We collected 100–283 images per syndrome for Angelman, Apert, Cornelia de Lange, Down, Fragile X, Progeria, Treacher Collins, or Williams-Beuren. Images were collected through publically available resources online accessible though search terms relating to each syndrome, primarily through support group pages and awareness event photographs. Source URLs were converted to shortened versions for the purposes of publication using TinyUrl (http://tinyurl.com/) (*Supplementary file 1*). The links provided are expected to decay with time and should only be considered exemplars of database composition. Images were captured through screen shots and saved as PNG or JPEG file formats.

The following two exclusion criteria were applied to the images: 1. The face needed to be clearly visible and oriented so that both eyes were visible. 2. The correct diagnosis was confirmed by an expert clinician (DRF). DRF inspected each image to validate the supposed syndrome diagnosis; images not validated were discarded. Variation in lighting, image resolution, pose, and occlusions has only been restricted when it obscures the facial characteristics (such as a hand covering the face). We also sought to avoid multiple images of the same individual at the same age in the database. No further restrictions were placed on variations in pose, facial expression, lighting, occlusions, or image quality.

In the same manner, smaller numbers of images were collected for Marfan, 22q11, Turner and Sotos syndromes (*Table 1*). Furthermore, we collected further published images of patients with confirmed genetic variants in genes of the RAS/MEK signaling pathways as well as in *PACS1* (*Rauen, 1993*, *2006*; *Bertola et al., 2007*; *Gripp et al., 2007*; *Makita et al., 2007*; *Rauen, 2007*; *Zampino et al., 2007*; *Nystrom et al., 2008*; *Schulz et al., 2008*; *Tidyman and Rauen, 2008*; *Cordeddu et al., 2009*; *Kratz et al., 2009*; *Zenker, 2009*; *Allanson et al., 2010*; *Wright and Kerr, 2010*; *Kleefstra et al., 2011*; *Lepri et al., 2011*; *Siegel et al., 2011*; *Schuurs-Hoeijmakers et al., 2012*; *Hopper et al., 2013*; *Twigg et al., 2013*).

3100 images were collected and manually annotated for training of the algorithms. Of these 2878 were successfully annotated by the automatic pipeline and are reported in the database counts of *Table 1*.

### Data and code availability

Original database, excluding the Gorlin collection, and previously published images (which are available from the cited original publications) can be requested by contacting CN (christoffer.nellaker@dpag.ox.ac.uk; Ferry Q, Steinberg J, Webber C, FitzPatrick DR, Ponting CP, Zisserman A, Nellåker C, 2014, Diagnostically relevant facial gestalt information from ordinary photos database). Requests will be assessed by a Data Access Committee (DAC) comprised of CPP, DRF, AZ, CN and Dr Zameel Cader of the Division of Clinical Neurology, University of Oxford. The DAC will make data available to researchers in good standing with the relevant institution and funding agencies (i.e., no known sanctions). The data are provided without copyright.

Pipeline code was written in python 2.7 and uses the module Ruffus (*Goodstadt, 2010*) for task management. The code is available through an open source MIT license at https://github.com/ChristofferNellaker/Clinical_Face_Phenotype_Space_Pipeline.

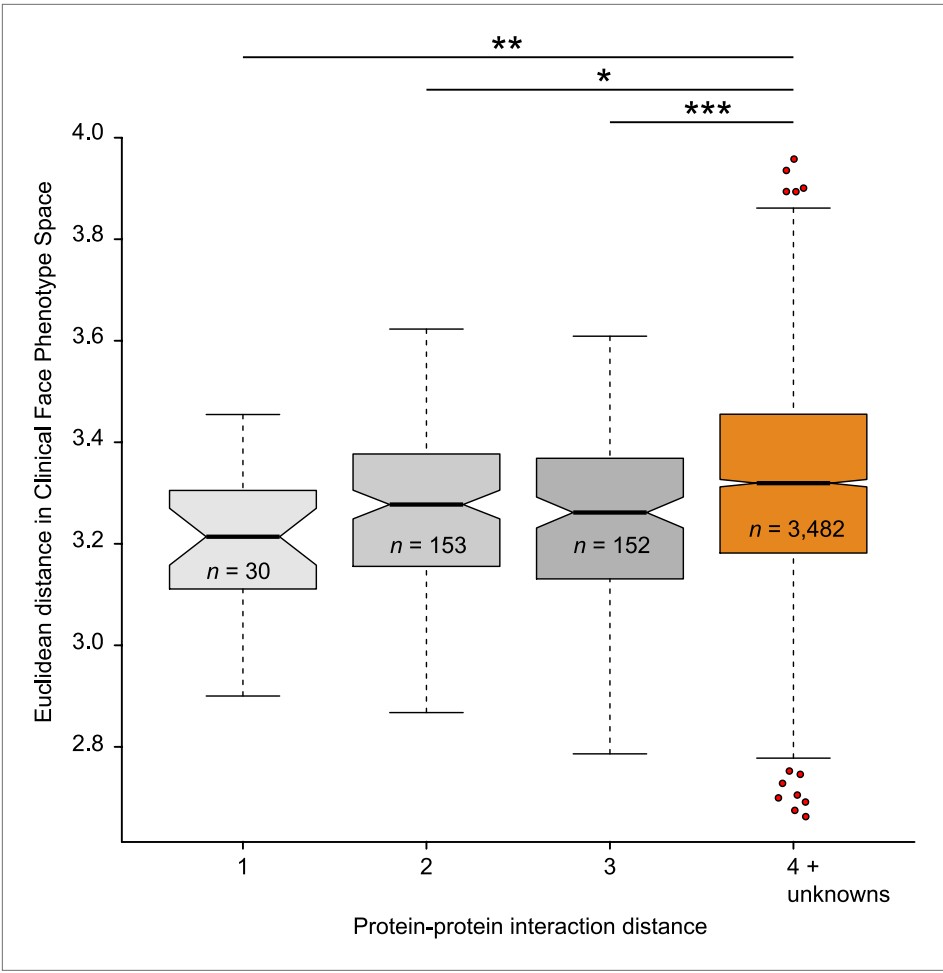

**Figure 5**. Clinical Face Phenotype Space recapitulates features of functional gene links between syndromes. Protein–protein interaction distances of 1–3 for genetically characterized syndromes are associated with significantly shorter Euclidean distance (arbitrary units) between syndromes in Clinical Face Phenotype Space as compared to syndromes with distance 4 or no known interaction distance (shown in orange) (Kruskal–Wallis tests with Bonferroni corrected p-values indicated as *p<0.05, **p<0.01, ***p<0.001). The Spearman correlation across all distances was r = 0.09, p<0.001. The numbers of pairwise syndrome comparisons underlying each of the interaction distances are listed within the respective boxes.

## Ethics statement

The manner and method by which images were collected from publically available sources and stored were acceptable research practices and do not require special consent from a Research Ethics Committee. Advice from legal services, research ethics board members and the Information Commissioner's Office (UK) was sought in arriving at this conclusion.

## Computer vision algorithm

The computer vision algorithm analyses a 2D photograph for the location of a face, annotates the facial landmark points, and extracts feature vectors for subsequent machine learning applications. MATLAB (MATLAB. R2011b Natick, Massachusetts: the MathWorks Inc.) with OpenCV (**Bradski, 2000**) was used to write scripts and functions for the algorithm. To identify a putative face in the photo, we used previously published algorithms (**Viola and Jones, 2001**). Within a box bounding the face, a pictorial structure model was used to identify 9 central facial feature points (**Everingham et al., 2009**), which then were used to initialize the placement of an additional 27 feature points. The resulting facial mesh structure was fitted to the image using Active Appearance Models (AAMs) (**Cootes et al., 1998**) to generate average face visualizations (**Figure 2—figure supplement 1**). The placement of the

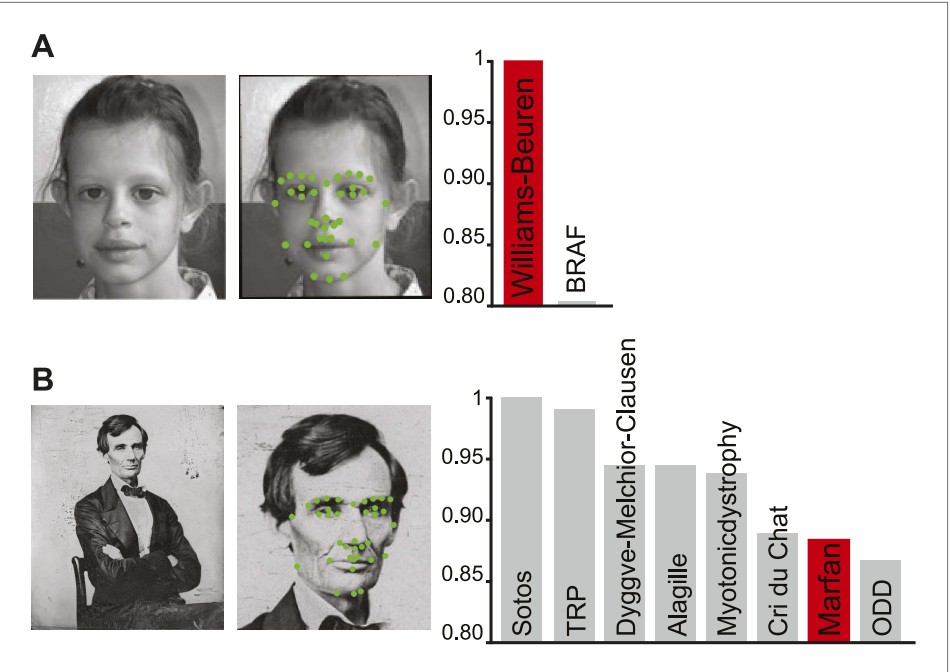

**Figure 6**. Class priority of diagnostic classifications for images. The full computer vision algorithm and Clinical Face Phenotype Space analysis procedure with diagnostic hypothesis generation exemplified by: (**A**) a patient (***Ferrero et al., 2007***) with Williams-Beuren (this patient figure was published in 'Giovanni Battista Ferrero, Elisa Biamino, Lorena Sorasio, Elena Banaudi, Licia Peruzzi, Serena Forzano, Ludovica Verdun di Cantogno, Margherita Cirillo Silengo. Presenting phenotype and clinical evaluation in a cohort of 22 Williams–Beuren syndrome patients. European Journal of Medical Genetics; 2007;50(5):327–337 Copyright 2007 Elsevier Masson SAS. All rights reserved'.). Note that panel **A** does not fall under a creative commons license and would need permissions from the copyright holders for future reproductions. (**B**) Abraham Lincoln. The former US President is thought to have had a marfanoid disorder, if not Marfan syndrome (***Gordon, 1962***; ***Sotos, 2012***). Bar graphs show class prioritization of diagnostic hypotheses determined by 20 nearest neighbors weighted by prevalence in the database. As expected, the classification of Marfan is not successfully assigned in the first instance as there were only 18 faces of individuals with Marfan in the database (making this an example of a difficult case with the current database). However, the seventh suggestion is Marfan, despite this being among 90 different syndromes and 2754 faces.

36 feature points was also further refined using custom written scripts based on consensus of exemplars (***Belhumeur et al., 2011***) (see Methods).

From the fitted constellation of facial landmarks two feature vectors were extracted. (1) The appearance as a concatenation of the pixel intensities of patches around the 9 inner facial feature points. (2) The shape vector was constructed as the normalized pairwise distances between all 36 facial feature points.

## Face detection

Each image was converted to JPEG and submitted to the Facial Detection (FD) module of the algorithm. Face detection was achieved using the OpenCV (***Bradski, 2000***) implementation of the Viola–Jones object detection framework (using Haar like features and a cascade of classifiers) (***Viola and Jones, 2001***). The output takes the form of a square bounding box delimiting the area of the picture where the face was found. Pictures containing the faces of healthy relatives (or others) were either discarded or cropped to only conserve regions with the patient face.

## Facial landmarks annotation

Manual annotation of the 36 feature points was performed on 3100 of the images in the image collection. These were used as the ground truth reference point for all subsequent training and test sets for evaluations of automated facial landmark annotation accuracies (***Figure 1—figure supplement 1***).

In the second step of the automatic algorithm detected faces were passed to a facial landmark annotation script (***Everingham et al., 2009***) (FLA module), which annotates the face with an initial set

of 9 well-characterized (salient) feature points. The 9 landmarks in that set were the medial and lateral canthi of the eyes, each subnasale, columella and the left and right vermilion border lateral midpoints. The FLA used the returned bounding box to approximate the location and size of the face to be annotated. Automatic annotation relies on a generative model of feature point position combined with a discriminative model of appearances. This joint model was based on the parts-based pictorial structure representation introduced by *Fischler and Elschlager (1973)*. For a given bounding box, the FLA module returns both a constellation of 9 landmarks and a corresponding confidence index computed via appearance mismatch with the model. We used this index to implement more robust, accurate and reliable annotation approaches (*Figure 1—figure supplement 1*) as described below.

Improved facial landmark annotation was performed with a custom script designed to refine the output from the FD-FLA modules in terms of annotation inaccuracies, false positive and false negative face detection. The images were transformed iteratively by mirror imaging, partial rotations (±45°), and by adding a frame around the image to produce 100 transformations of the original. Each image was subsequently submitted through the FD -FLA modules and returned a constellation of 9 points with associated confidence scores recorded. A consensus map is constructed by confidence weighted averaging of high confidence feature annotations thus reducing the number of spurious annotations and increasing annotation accuracy (*Figure 1—figure supplement 1*).

From the improved 9 facial feature points, we expanded the feature detection to 36 feature points encompassing the points indicated in *Figure 1—figure supplement 1*.

## Consensus of exemplar

We developed a computational module inspired by *Belhumeur et al. (2011)* to determine the localization of the 36 landmarks. Consensus of exemplar (CoE) relies on part base classifiers used to localize potential candidate points and a database of face exemplars used to introduce a shape prior in the search for the best constellation.

While only a constellation of 9 feature points (C9) is required to compute the appearance feature vector, the shape feature vector relies on a constellation of 36 feature points (C36) covering the inner face in greater details along with its outlined (Jaw line). Anatomical landmarks covered in C36 are shown in *Figure 1—figure supplement 1*. We used the C9 obtained via the improved FLA module (see previous section) to initiate the automatic search for C36.

For each of the facial feature points in C36, we delimited a region of interest (ROI) for the algorithm to consider using the following heuristic: 1000 exemplar faces are sample from among the controls in the database (*Table 1*). For each exemplar face i, we registered the $C9_i$ to the C9 of our query face using Procrustes algorithm. Next the sum of squares error was used to sort exemplars in order of accuracy with which $C9_i$ registered to C9. The top 20 exemplars were used to map their $C36_i$ to the query face using the transform $T_i$ obtained during registration. A consensus C36 for the query face was then derived by averaging all registered $C36_i$. Finally, for each feature point in C36, we defined a square ROI centered on the consensus point with two palpebral fissures (PF) length dimensions. The PF length was the average between right and left eyes and estimated based on C9.

The final C36 was derived from the ROIs by using a combination of part based detectors and a consensus of shape exemplars. Support Vector Machine (SVM) classifiers (using Gaussian kernels) were trained to recognize each feature point in C36. We used a database of manually annotated control patients to obtain positive and negative training sets for each part classifier. The feature vector associated with a particular feature point was obtained by cropping a square patch centered on this point and describing its pixel content with a pyramidal histogram of gradients (PHOG).

Going back to the query face, each of the 36 ROIs was submitted to the corresponding part detector. From this we obtained a set of 20 potential candidates (PC) for each part within the ROI (where candidates were sorted based on the classifier decision values). Next, we randomly sampled exemplars of C36 from the control database and registered them to the query face in order to enforce a shape prior. To avoid spurious outlier PC to drive the registration off, we randomly select PC to represent three randomly selected points from C36. Exemplar $C36_i$ were registered via Procrustes algorithm to the query face using only these three PC. The registered $C36_i$ were scored by submitting its feature points to the part classifiers. We retained the top 20 $C36_i$. Finally, each feature point of the final C36 for the query was derived independently as a consensus between probability density maps based on the PCs and the classifier decision values over the corresponding ROI.

## Appearance descriptor feature vector

Using the set of 9 inner points annotated to the face by the FLA module and refined by the CoE module, four additional points were generated: left and right center of the eye, nasion (between the eyes) and center of the mouth. The points were then used to register the face, via an affine transformation T, to a canonical set of corresponding points (face shape template). Next, circular patches were generated around the canonical points which were mapped back to the original picture using the inverse of T. This process creates 13 ellipses which were then used to crop the image content by bilinear interpolation. Extraction of the patch was performed as in *Everingham et al. (2009)*.

The appearance feature vector was obtained by a concatenation of the pixel gray-scale content of the 13 cropped patches (size of the feature vector = 1937). While patch content could be further processed before concatenation (gradients, HOG, PHOG, SIFT, Gabor Filter, Local binary Pattern, etc) the gain in terms of discriminative power and relevant feature extracted was negligible compared to the computational cost in terms of memory consumption.

## Shape descriptor feature vector

The features vector was built from the constellation of the 36 landmarks annotations from the CoE module. We described the face shape as a vector d, the set of pair-wise distances through the constellation, resulting in a feature vector with 630 elements.

To compare the different constellations, each was registered via Procrustes transformation to the average constellation of the AAM model (canonical face mesh, see below). The vector of pair-wise distances was then normalized so that any distance variations were relative to the corresponding distances measured on the average control face template.

## Average faces and morphs using Active Appearance Model

Introduced in 1998 by *Cootes et al. (1998)*, the active appearance model (AAM) was designed to identify a set of facial landmarks on a given face. This task is achieved by iteratively modifying both the shape and the position (location) of a structured face mesh in which nodes represent target landmarks.

We constructed a training set by manual annotation of 3100 patient images with the 36 landmarks (*Figure 1—figure supplement 1*). All constellations were registered together using an iterative Procrustes algorithm. We computed the average constellation and used it as a reference with which to build a canonical face mesh via Delaunay triangulation. Based on the obtained triangulation, we generated a face mesh dividing each patient's face into sub-regions. Using piecewise affine warping, we independently mapped the pixel content of each sub-region to the corresponding triangle in the canonical face mesh. We thus obtained a registered version of each patient's facial appearance. Shapes and appearance registrations were used in a principal component analysis (PCA) to generate both the shape and appearance models for use with the AAM.

Using the AAM statistical models of shape and appearance that derive from the training of the AAM, we created a visual representation of canonical traits/phenotypes. Average faces were created from the images of patient with the same genetic disorder. AAM models were generated for each group, and after registration of the constellations (annotation) to each we derived the average shape constellation. A face mesh was generated from these constellations by triangulation (Delaunay) and the appearance of each individual was mapped to the average face mesh by piecewise affine warping. Thus, this average face mesh recapitulated the canonical phenotype of the syndromes.

Furthermore, we created morphs between control and syndrome faces by registering appearance to the average shape of the controls (*Figure 2—figure supplement 1*). By computing the average across the syndrome group and the control group one can obtain the average appearance of the syndrome $A_{0s}$ and control $A_{0c}$. Given that both $A_{0s}$ and $A_{0c}$ have identical dimensionality, we morphed appearance from one to the other with appearance for frame k where $A_k = A_{0s} + k/K*(A_{0c} - A_{0s})$ and where K is the total number of frames. Similarly, we considered the average face mesh of the syndrome group $M_{0s}$ and the control group $M_{0c}$ to obtain the face mesh at frame k as $M_k = M_{0s} + k/K* (M_{0c} - M_{0s})$. Finally, for each frame k, both shape (face mesh) and appearance were combined by piecewise affine warping of $A_k$ to $M_k$.

## Support Vector Machine classification

We used SVMs to fine tune both appearance and shape descriptors. Differences in binary classification accuracies allow us to infer relative feature delineation capacity by our descriptors. We employed

Libsvm to perform classification. The accuracies based on the raw feature vectors were comparable to accuracies reported in previous studies (*Boehringer et al., 2006*, *2011*).

## Binary classification

SVM classifiers were trained on both shape and appearance feature vectors separately for each 36 pairwise combination of control and 8 syndromes (*Table 1*). Each binary classification was repeated 10 times with randomly generated positive/negative training and test sets with a 4:1 ratio. The linear kernel SVM classification accuracies using the final shape and appearance descriptors are shown in *Figure 4—figure supplement 1*. We estimate a total accuracy when fusing information from shape and appearance feature vectors based SVMs by summing decision values returned by the shape and appearance classifiers respectively.

In all classification experiments, we used both original and mirror versions of each image. Given a binary experiment to distinguish group G1 from G2 (where G refers to a set of images of a syndrome), we randomly partitioned the sets into training and test sets: $G1_{tr}$, $G1_{te}$, $G2_{tr}$, and $G2_{te}$ respectively. From these we generated the corresponding mirror image sets: $G1_{tr}^m$, $G1_{te}^m$, $G2_{tr}^m$, and $G2_{te}^m$. The SVM model was trained using $G1_{tr} + G1_{tr}^m$ as the positive set and $G2_{tr} + G2_{tr}^m$ as the negative set. Decision values were tuned to give equal error rates (the number of false positives = number of false negatives). With the trained SVM models, we next submitted the sets $G1_{te} + G2_{te}$ and $G1_{te}^m + G2_{te}^m$ separately to the classifier. Thus, for each instance in the test set we obtained two decision values. The final classification was determined by the sum of the decision values.

## Forced choice classification

We next applied the classification problem to assigning a face as being one of the 8 syndromes (*Table 1*). Each of the 8 syndrome groups was randomly split into training and test sets with a ratio of 4:1. We used the training sets to train a linear binary SVM classifier for each of the 28 pairs. Each image in the training set was submitted to all 28 classifiers. The decision value was returned by each classified and used as a probabilistic estimate P for the test instance to belong to the positive class. Thus, after presenting instance *i* to the binary classifier distinguishing syndrome *j* (positive) from syndrome *k* (negative), we assigned *i* a vote of weight P for syndrome *j* and a vote of weight 1-P for syndrome *k*. After summation of the votes from the 28 classifiers, the instance *i* was labeled as belonging to the syndrome diagnosis with the highest probability.

The confusion matrix averaged from 10 repeats of the forced choice experiment is shown in *Figure 4—figure supplement 2*.

## Clinical Face Phenotype Space construction

We performed PCA on the shape and appearance feature vectors to reduce dimensionality from 2567 to a concatenation of 340 orthogonal vectors. This was then used to transform the feature space with Large Margin Nearest Neighbor (LMNN, *Weinberger and Saul, 2009*). LMNN is an optimization algorithm that uses a training set of pairs of vector labels ($x_i$; $l_i$) and learns a Mahalanobis distance that maximizes the kNN classification over a training set. Note that even if the system only considers local information (i.e., number of intruders for each instance) the final metric is global. The Mahalanobis distance was computed as $dist(x_i; x_j) = (x_i - x_j)^t L^t L (x_i - x_j)$ where L is a linear matrix. It is equivalent to the Euclidean distance taken in the space after transformation by L. That is to say, LMNN linearly transforms dimensions in feature space to maximize the margins separating classes of labeled instances. This should, in principle expand dimensions with phenotypically relevant information and compresses dimensions uninformative for classification.

To validate the characteristics of the transformation of spurious and phenotypic vectors in Clinical Face Phenotype Space, we performed a series of experiments based on projected 3D faces. We used the 3D facial model proposed by Blanz & Vetter (*Blanz, 2006*) http://faces.cs.unibas.ch/bfm/ which allowed us to create faces with a direct control over shape, appearance, lighting, and facial pose. We synthesized 5 test faces at random moving along the first 15 components of both the shape and appearance models. For each face, we generated a set of 20 images for each combination of 5 head rotations and 4 lighting conditions. We use these simulated images to compare similarity measures in the raw feature space and in Clinical Face Phenotype Space (*Figure 3*, *Figure 1—figure supplement 2*). We performed a reorientation of the raw feature vectors and Clinical Face Phenotype Space using PCA without dimensionality reduction in order to sort the dimensions by variation magnitude. This allowed us to assess the relative contributions of phenotypic variation and spurious variations to

clustering of faces. The strongest influences on clustering would be expected to be encoded in the first modes of variation. Placing the synthetic faces in the reoriented spaces allowed us to describe the PCA signatures of phenotypic variations (shape, appearance) and spurious variation (lighting, head pose).

## Clinical Face Phenotype Space validation and visualization

We used several dimensionality reduction methods and metrics for visualization and estimation of the properties of Clinical Face Phenotype Space. We developed an estimate of search space reduction to determine the improvements in clustering in Clinical Face Phenotype Space controlling for the composition of the database. Essentially, this calculates the degree to which intruders in a nearest neighbor search between instances of the same syndrome are excluded in Clinical Face Phenotype Space. This equates to a factor estimate of increased clustering, CIF (details of the procedure are provided below).

We used a 20 nearest neighbor linkage map to visualize Clinical Face Phenotype Space using force directed graphs implemented through Gephi (*Bastian et al., 2009*).

Protein–protein interaction data were obtained from DAPPLE (*Rossin et al., 2011*). After conversion to Ensembl gene IDs, 126,586 interactions between 10,442 genes remained. We considered the data as a network of genes, with edges denoting an interaction. The shortest paths between two genes were computed using Dijkstra's algorithm (*Dijkstra, 1959*). We calculated the median pairwise Euclidean distance between syndromes in Clinical Face Phenotype Space. The correlation between these two data sets underlies *Figure 5*. Clinical Face Phenotype Space distance between groups was tested using Kruskal–Wallis (*Kruskal and Wallis, 1952*) tests with Bonferroni (*Bonferroni, 1935*, *1936*) multiple testing correction.

## Estimating improvements in clustering

Next, we performed estimations of clustering of syndromes in face space. Initial tests using kNN-classifiers showed that the classification accuracies were heavily dependent on spread and cardinality of the syndrome in the database. We went on to develop an estimate of search space reduction, hereafter referred to Clustering Improvement Factor (CIF), to determine the improvements in clustering in Clinical Face Phenotype Space controlling for the composition of the database (a simulated example is provided in *Figure 4—figure supplement 3*).

We considered a syndrome with $N_p$ positive and $N_n$ negative instances in the Clinical Face Phenotype Space. We defined the CIF as

$$\text{CIF} = \frac{\text{expected rank (r) of nearest positive match under random ranking}}{\text{observed average rank (r) of nearest positive match}} = \frac{E(r)}{O(r)}$$

with the average taken across all instances of the syndrome. $O(r)$ was calculated from the observations in the Clinical Face Phenotype Space. To compute $E(r)$, we used probability theory as follows.

Under a random ranking for a given positive query, the other $N_p-1$ positive instances are each placed independently among the $N_n$ negative instances, with a uniform discrete probability distribution. We defined the random variable $N_i$ as the number of negative instances ranked higher than the first positive instance, so $N_i$ takes integer values $0 \le N_i \le N_n$.

For a given positive query, the expected rank of the nearest positive match is the expected value of $N_i+1$, denoted by $E(N_i)+1$. To calculate $E(N_i)$, we used the definition of expectation:

$$E(N_i) = \sum_{j=0}^{N_n} j \Pr(N_i = j)$$

since $N_i$ can only take non-negative integer values, for each possible value j between 0 and $N_n$,

$$\Pr(N_i = j) = \Pr(N_i \ge j) - \Pr(N_i \ge j + 1)$$

Substituting this in the formula for $E(N_i)$,

$$E(N_i) = \sum_{j=0}^{N_n} j \big[\Pr(N_i \ge j) - \Pr(N_i \ge j+1)\big].$$

Rewriting the sum,

$$E(N_i) = \sum_{j=1}^{N_n} \big[j\Pr(N_i \ge j) - (j-1)\Pr(N_i \ge j)\big] = \sum_{j=1}^{N_n} \Pr(N_i \ge j)$$

For a given number j, $Pr(N_i \geq j)$ is the probability that all positive instances were placed after j negative instances. For any given individual positive instance, such placement has probability $1 - \dfrac{j}{N_n + 1}$.

Since placement of all positive instances is independent, this gives $Pr(N_i \geq j) = \left(1 - \dfrac{j}{N_n + 1}\right)^{N_p - 1}$.

Therefore, $E(N_i) = \sum_{j=1}^{N_n} Pr(N_i \geq j) = \sum_{j=1}^{N_n} \left(1 - \dfrac{j}{N_n + 1}\right)^{N_p - 1}$

Finally, this gives $E(r) = 1 + E(N_i) = 1 + \sum_{j=1}^{N_n} \left(1 - \dfrac{j}{N_n + 1}\right)^{N_p - 1}$.

## Querying Clinical Face Phenotype Space

We developed two methods to retrieve information about the neighborhood of a given face placed in the Clinical Face Phenotype Space. Firstly, we assigned a syndrome classification based on the identity of its k nearest neighbors in Clinical Face Phenotype Space. Based on the neighbors' labels a list of syndromes to which the new face could belong was created. The number of neighbors supporting each hypothesis was compared with the probability to see N instances of that syndrome when sampling k from the population of faces in Clinical Face Phenotype Space.

Secondly, we estimate the relative similarity between specific faces given the density of points in a local region of Clinical Face Phenotype Space. This is calculated as $p0p1 = d_{0,1}/\sqrt{d_0 d_1}$, where $d_{0,1}$ is the similarity measure between the query and its neighbor, $d_0$ is the average of similarities between the query and k = 20 neighbors and $d_1$ is the average of similarities between the neighbor of the query and k of the neighbor's neighbors. *Figure 4—figure supplement 4* illustrates the method and metrics using a simulated example.

We see Clinical Face Phenotype Space as a means to facilitate collaborative investigations of genetic diseases between clinicians. Of course, sharing of data raises questions regarding ethics approval and data security. These questions are tightly linked to the debate of how clinical sequencing information should be treated in global health care systems. We anticipate that it would be suitable for future implementations of Clinical Face Phenotype Space to follow similar guidelines as for clinical sequencing data.

## Acknowledgements

We would like to thank Dr Zameel Cader for advice at the initiation of this project. Also, we would like to thank Professor Raoul MC Hennekam for kindly allowing us access to the Gorlin collection. We are grateful to Dr Bronwyn Kerr for suggesting publication with RAS/MEK mutation patient images. Author contributions: CN, AZ, CPP, CW, and DRF conceived the study. QF collected the database and annotated the images. QF and CN wrote the code. QF, AZ, JS, and CN performed the analyses. DRF validated the syndrome classification of images. CN drafted the manuscript. All co-authors edited and approved the final manuscript.

## Additional information

### Competing interests

CPP: Senior editor, *eLife*. The other authors declare that no competing interests exist.

### Funding

| Funder | Grant reference number | Author |
|---|---|---|
| Medical Research Council (MRC) | | Caleb Webber, David R FitzPatrick, Chris P Ponting |
| Wellcome Trust | | Julia Steinberg |
| European Research Council | 228180 VisRec | Quentin Ferry, Andrew Zisserman |
| Oxford Biomedical Research Centre | | Christoffer Nellåker |
| Medical Research Council (MRC) | Centenary Award | Quentin Ferry, Christoffer Nellåker |

The funders had no role in study design, data collection and interpretation, or the decision to submit the work for publication.

## Author contributions
QF, Collected the database, annotated the images, wrote the code and performed analyses, edited and revised the manuscript., Acquisition of data, Analysis and interpretation of data, Drafting or revising the article; JS, Performed analyses and edited the manuscript., Analysis and interpretation of data, Drafting or revising the article; CW, Conceived the study and edited the manuscript., Conception and design; DRFP, Conceived the study, validated the syndrome classification of images and edited the manuscript., Conception and design, Acquisition of data, Drafting or revising the article; CPP, Conceived the study and edited the manuscript., Conception and design, Drafting or revising the article; AZ, Conceived the study, performed data analysis and edited the manuscript., Conception and design, Analysis and interpretation of data, Drafting or revising the article; CN, Conceived the study, wrote code, performed analyses and drafted the manuscript., Conception and design, Analysis and interpretation of data, Drafting or revising the article

## Ethics
Human subjects: The manner and method by which images were collected from publically available sources and stored were acceptable research practices and do not require special consent from a Research Ethics Committee. Advice from legal services, research ethics board members and the Information Commissioner's Office (UK) was sought in arriving at this conclusion.

# Additional files

## Supplementary file
• Supplementary file 1. Tinyurl links to sources for the database. Prefix the 7 characters with http://tinyurl.com/. Links are expected to decay with time; the full dataset will be released to researchers at the discretion of a Data Access Committee.

## Major dataset
The following dataset was generated:

| Author(s) | Year | Dataset title | Dataset ID and/or URL | Database, license, and accessibility information |
|---|---|---|---|---|
| Ferry Q, Steinberg J, Webber C, FitzPatrick DR, Ponting CP, Zisserman A, Nellåker C | 2014 | Diagnostically relevant facial gestalt information from ordinary photos database. | Original database, excluding the Gorlin collection, and previously published images (which are available from the cited original publications) can be requested by contacting CN (christoffer.nellaker@dpag.ox.ac.uk). Requests will be assessed by a Data Access Committee (DAC) comprised of CPP, DRF, AZ, CN and Dr Zameel Cader of the Division of Clinical Neurology, University of Oxford. The DAC will make data available to researchers in good standing with the relevant institution and funding agencies (i.e., no known sanctions). The data are provided without copyright. | |

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
