## [Decision Letter]

Thank you for sending your work entitled “Diagnostically-relevant facial gestalt information from ordinary photos” for consideration at *eLife*. Your article has been favorably evaluated by a Senior editor, Detlef Weigel, a Reviewing editor, and 2 reviewers, one of whom, Peter Claes, has agreed to reveal his identity.

The Reviewing editor and the reviewers discussed their comments before we reached this decision, and the Reviewing editor has assembled the following comments to help you prepare a revised submission.

With genome sequencing having become cheaper and cheaper over the past few years, the bottleneck in genetics is increasingly shifting to accurate, inexpensive phenotyping. Even for Mendelian genetic disorders, misdiagnosis rates are shockingly high. A sizeable fraction of genetic disorders is associated with craniofacial abnormalities, and the approach presented in this study greatly facilitates the acquisition and analysis of craniofacial information from simple photographs, as opposed to more complex 3D scans that are not necessarily suitable for every day medical diagnostics. The reviewers felt that the work was of high quality, and they had only some minor comments that need to be addressed in the revision.

*From Reviewer 1:*

The simple term “Face-Space” for the new representation presented by the authors is confusing. When doing a simple search on the internet, the concept of a Face-Space is well known as a space coding for facial shape variations due to general differences like differences in identity and not necessary clinical phenotype. In face recognition and perception, the term face-space is already widely used and does not reflect the same space as presented by the authors. I suggest defining the new representation differently to avoid confusion, for example Diagnostic Face Space.

In the Introduction the advantages over current studies using 3D images is slightly exaggerated. The single most important, BUT major advantage of the proposed approach is the easy access to ordinary images in contrast to expensive 3D acquisition systems. Rare genetic diseases are hard to find and collect, and through the use of ordinary 2D images one manages to acquire a substantial amount of samples across the world (which is typically required in AI based image strategies). In 3D, pose and illumination variations do not matter as such (which is well-documented in the literature of 2D versus 3D face recognition [1]) because pose differences can be normalized and syndromic diagnosis is mainly done on shape (and not texture) only. In contrast 2D images suffer a lot from pose and illumination differences, and the use of LMNN is a clever way of reducing their effect in clustering for diagnostic purposes as proposed by the authors (as also shown by [61] on 2D face recognition data). The easy access to 2D images by itself is a strong argument in favor of the proposed approach.

It is intriguing to see the generalization capabilities of the “Face-Space”. Trained on only 8 syndromes, it is capable of being applied to a much wider range of syndromes.

With regards to the SVM classification on the raw feature space, it is unclear to me, why this is performed. It seems to be a 2D implementation of a syndrome classification comparable to the 3D based classification done in the work of Hammond et al. The way I can read this in the context of the remainder of the paper is: In essence a traditional classification in the raw feature space using SVM increased from 94.4%, to 99.5% in “Face-Space” using kNN classification. However, I'm not sure if this is correct or not.

What is the stability of the main axes in “Face-Space”? In other words, do the distance metrics change a lot when using 8 different syndrome classes than the ones used for training in the manuscript? Did the authors experiment along these lines or not?

Finally the clustering improvement factor is novel and seems like an interesting metric proposed by the authors. Due to its novelty it is hard to give an interpretation to the metric and a more elaborated sample computation of the metric using the database composition as presented in the manuscript would be helpful in the methods section.

*From Reviewer 2:*

There are a couple of areas where the authors could add to the paper: One concerns the nature of the appearance model. We know that the faces show quite significant rigid pose variations. Would the model be improved by explicit modelling of the 3D variation? Perhaps an approach similar to that adopted by Iain Matthews and collaborators, building simultaneous 2D and 3D models might be worthwhile.

The other area relates to alternatives to the LMNN transform. Did the authors consider any other transforms? Do they rely on fundamentally different features of the data? Do they have significantly different performances?

I do not believe that either of these areas require additional computation – really just a short discussion to show the reader that there are alternative techniques within this area and choices have to be made about the computational and modelling techniques.

---

## [Author Response]

From Reviewer 1:

*The simple term “Face-Space” for the new representation presented by the authors is confusing. When doing a simple search on the internet, the concept of a Face-Space is well known as a space coding for facial shape variations due to general differences like differences in identity and not necessary clinical phenotype. In face recognition and perception, the term face-space is already widely used and does not reflect the same space as presented by the authors. I suggest defining the new representation differently to avoid confusion, for example Diagnostic Face Space*.

We agree. The term “Face Space” has now been replaced throughout with the more specific term “Clinical Face Phenotype Space”. This better reflects the character and purpose of the algorithms.

*In the Introduction the advantages over current studies using 3D images is slightly exaggerated. The single most important, BUT major advantage of the proposed approach is the easy access to ordinary images in contrast to expensive 3D acquisition systems. Rare genetic diseases are hard to find and collect, and through the use of ordinary 2D images one manages to acquire a substantial amount of samples across the world (which is typically required in AI based image strategies). In 3D, pose and illumination variations do not matter as such (which is well-documented in the literature of 2D versus 3D face recognition [1]) because pose differences can be normalized and syndromic diagnosis is mainly done on shape (and not texture) only. In contrast 2D images suffer a lot from pose and illumination differences, and the use of LMNN is a clever way of reducing their effect in clustering for diagnostic purposes as proposed by the authors (as also shown by*
[61]
*on 2D face recognition data). The easy access to 2D images by itself is a strong argument in favor of the proposed approach*.

Comparisons to 3D imaging to the current approach, and previous work in 2D imaging to the current approach, are now separated into two sentences to provide greater clarity.

“While 3D imaging studies have shown high discriminatory power in terms of classification they have relied on specialized imaging equipment and patient cooperation. Previous work with 2D images has relied on manual annotation of images, controlling lighting, pose and expression to allow consistent analyses. These factors greatly limit the availability, and ultimately the potential widespread clinical utility of such approaches.”

*It is intriguing to see the generalization capabilities of the “Face-Space”. Trained on only 8 syndromes, it is capable of being applied to a much wider range of syndromes*.

*With regards to the SVM classification on the raw feature space, it is unclear to me, why this is performed. It seems to be a 2D implementation of a syndrome classification comparable to the 3D based classification done in the work of Hammond et al. The way I can read this in the context of the remainder of the paper is: In essence a traditional classification in the raw feature space using SVM increased from 94.4%, to 99.5% in “Face-Space” using kNN classification. However, I'm not sure if this is correct or not*.

The SVM work is now better contextualised to clarify the purpose of these investigations. The following sentences are now amended:

“We next sought to compare the syndrome relevant information content of the feature descriptors to previous studies (27, 28, 11, 60). We found that classification analysis based on support vector machines provided similar accuracies to previous work, despite disparities in image variability (average classification accuracy 94.4%, see Figure 4—figure supplement 1, Figure 4—figure supplement 2 and Methods).”

*What is the stability of the main axes in “Face-Space”? In other words*, *do the distance metrics change a lot when using 8 different syndrome classes than the ones used for training in the manuscript? Did the authors experiment along these lines or not?*

To address this, we performed a test using leave-one-out training of the Clinical Face Phenotype Space. However, beyond affirming the expected loss in fidelity in classifications within the resulting space, we were unable to find a suitable way to compare the compositions of the primary axes.

*Finally the clustering improvement factor is novel and seems like an interesting metric proposed by the authors. Due to its novelty it is hard to give an interpretation to the metric and a more elaborated sample computation of the metric using the database composition as presented in the manuscript would be helpful in the methods section*.

We agree that the Clustering Improvement Factor (CIF) metric needed to be discussed in greater detail in the manuscript.

We have taken time to further validate and explain the CIF metric, and have improved this metric by better modelling of the background expectation. The consequence is that the average CIF for the 8 syndromes is 13.7 fold (note: later refined to 27.6-fold) over random chance. A CIF value of 1 is random chance (Figure 4). Note that Figure 3 and Figure 4 have been updated with the new CIF estimates.

To improve the explanation we have added a toy example of the CIF calculation as Figure 4—figure supplement 3.

The CIF explanation in the Methods section has been extended:

“Next we performed estimations of clustering of syndromes in face space. Initial tests using kNN classifiers showed that the classification accuracies were heavily dependent on spread and cardinality of the syndrome in the database. We went on to develop an estimate of search space reduction, hereafter referred to Clustering Improvement Factor (CIF), to determine the improvements in clustering in Clinical Face Phenotype Space controlling for the composition of the database (a simulated example is provided in Figure 4—figure supplement 3). We considered a syndrome with Np positive and Nn negative instances in the Clinical Face Phenotype Space. We defined the CIF as:CIF=expected rank (r) of nearest positive match under random rankingobserved average rank (r) of nearest positive match=E(r)O(r)

with the average taken across all instances of the syndrome. O(r) was calculated from the observations in the Clinical Face Phenotype Space. To compute E(r), we used probability theory as follows.

Under a random ranking for a given positive query, the other Np-1 positive instances are each placed independently among the Nn negative instances, with a uniform discrete probability distribution. We defined the random variable Ni as the number of negative instances ranked higher than the first positive instance, so Ni takes integer values 0≤Ni≤Nn.

For a given positive query, the expected rank of the nearest positive match is the expected value of Ni+1, denoted by E(Ni)+1. To calculate E(Ni), we used the definition of expectation: E(Ni)=∑j=0NnjPr(Ni=j)

Since Ni can only take non-negative integer values, for each possible value j between 0 and Nn,

Pr(Ni=j)=Pr(Ni≥j)−Pr(Ni≥j+1)

Substituting this in the formula for E(Ni), E(Ni)=∑j=0Nnj[Pr(Ni≥j)−Pr(Ni≥j+1)]

Rewriting the sum, E(Ni)=∑j=1Nn[jPr(Ni≥j)−(j−1)Pr(Ni≥j)]=∑j=1NnPr(Ni≥j)

For a given number j, Pr(Ni≥j) is the probability that all positive instances were placed after j negative instances. For any given individual positive instance, such placement has probability 1−jNn+1. Since placement of all positive instances is independent, this gives Pr(Ni≥j)=(1−jNn+1)Np−1.

Therefore, E(Ni)=∑j=1NnPr(Ni≥j)=∑j=1Nn(1−jNn+1)Np−1

Finally, this gives E(r)=1+E(Ni)=1+∑j=1Nn(1−jNn+1)Np−1.”

From Reviewer 2:

*There are a couple of areas where the authors could add to the paper:*

*One concerns the nature of the appearance model. We know that the faces show quite significant rigid pose variations. Would the model be improved by explicit modelling of the 3D variation? Perhaps an approach similar to that adopted by Iain Matthews and collaborators, building simultaneous 2D and 3D models might be worthwhile*.

Please see response below.

*The other area relates to alternatives to the LMNN transform*. *Did the authors consider any other transforms? Do they rely on fundamentally different features of the data? Do they have significantly different performances?*

Please see response below.

*I do not believe that either of these areas require additional computation – really just a short discussion to show the reader that there are alternative techniques within this area and choices have to be made about the computational and modelling techniques*.

Thank you for these comments. Indeed an explicit modelling of the 3D variation is an avenue we are pursuing in follow up research. We were not able to explore other metric learning alternatives within the scope of this pilot study but are pursuing this line of questioning in our current research. These points are now discussed briefly in the Discussion:

“Among the approaches that will be tested in future works are: increasing the number of feature points across the cranium, using profile images and taking advantage of multiple images of the same individual. Furthermore we will be exploring performing explicit modelling of the 3D variation for 2D images (45), other types of feature descriptors, alternative metric learning and dimensionality reduction approaches (54).”